# *Bifidobacterium infantis*-Mediated Herpes Simplex Virus-TK/Ganciclovir Treatment Inhibits Cancer Metastasis in Mouse Model

**DOI:** 10.3390/ijms241411721

**Published:** 2023-07-21

**Authors:** Changdong Wang, Yanxi Shen, Yongping Ma

**Affiliations:** Department of Biochemistry & Molecular Biology, Molecular Medicine & Cancer Research Center, College of Basic Medicine, Chongqing Medical University, Chongqing 400016, China

**Keywords:** tumor metastasis, *Bifidobacterium*, thymidine kinase, ganciclovir, gastric cancer cell, bioinformatics

## Abstract

Previous studies have found that *Bifidobacterium infantis*-mediated herpes simplex virus-TK/ganciclovir (BF-TK/GCV) reduces the expression of VEGF and CD146, implying tumor metastasis inhibition. However, the mechanism by which BF-TK/GCV inhibits tumor metastasis is not fully studied. Here, we comprehensively identified and quantified protein expression profiling for the first time in gastric cancer (GC) cells MKN−45 upon BF-TK/GCV treatment using quantitative proteomics. A total of 159 and 72 differential expression proteins (DEPs) were significantly changed in the BF-TK/GCV/BF-TK and BF-TK/GCV/BF/GCV comparative analysis. Kyoto encyclopedia of genes and genomes (KEGG) pathway analysis enriched some metastasis-related pathways such as gap junction and cell adhesion molecules pathways. Moreover, the transwell assay proved that BF-TK/GCV inhibited the invasion and migration of tumor cells. Furthermore, immunohistochemistry (IHC) demonstrated that BF-TK/GCV reduced the expression of HIF−1α, mTOR, NF-κB1-p105, VCAM1, MMP13, CXCL12, ATG16, and CEBPB, which were associated with tumor metastasis. In summary, BF-TK/GCV inhibited tumor metastasis, which deepened and expanded the understanding of the antitumor mechanism of BF-TK/GCV.

## 1. Introduction

Tumor metastasis is the main cause of poor comprehensive treatment and survival prognosis [1,2]. Despite recent therapeutic advances in cancer treatment, many data show that more than 90% of tumor patients die of distal target organ metastasis [3]. Unlike primary tumors, which can often be cured using local surgery or radiation, metastatic cancers are largely non-curable for circulating tumor cells [1,2,4]. Therefore, the inhibition of tumor metastasis is still a challenge in current antitumor research.

To reduce the high relapse rate after radical surgery and conventional therapies (radiotherapy, chemotherapy, and immunotherapy) [5], novel approaches such as cancer gene therapy have raised hope that the survival rate of tumor patients can be significantly improved [6,7,8,9]. Some gene therapy products have been approved for clinical use, for example, recombinant adenovirus-p53 [10]. Of the gene delivery systems, suicide gene therapy is the most frequently used method for solid tumors, which is used to eliminate tumor cells in three ways: the direct killing effect, bystander effect, and cell signaling pathway [11,12]. The herpes simplex virus thymidine kinase (TK) gene combined with ganciclovir (GCV) is one of the systems that has been studied in the most depth [13,14,15,16,17,18,19]. In this system, viral TK is expressed and simultaneously metabolizes the prodrug GCV to mono-phosphorylated GCV, which will be further converted into GCV triphosphate GCV, an analogue of deoxyguanosine triphosphate. Consequently, triphosphate GCV inhibits DNA synthesis and leads to tumor cell death [18,20].

A safe, effective, and controllable gene delivery system is important for gene therapy. There are three main types of vectors: viral vectors (adenoviral, adeno-associated viral), non-viral vectors (polymers, liposomes), and bacterial vectors (*Bifidobacterium*, *Escherichia coli*, and *Salmonella*) [21,22,23]. Previous studies have demonstrated that some obligate anaerobes (*Bifidobacterium*, *Clostridium*) or facultative anaerobes (*Salmonella*, *Escherichia coli*, *Listeria monocytogenes*) can selectively accumulate and proliferate within tumors and suppress tumor growth [24,25]. However, fundamental issues such as safety and targeting remain to be resolved before engineered *Salmonella* can be used in the clinic [24]. *S. typhimurium* mutant VNP20009 and its derivative strain TAPET-CD have even been applied in human clinical trials [26,27]. Furthermore, *Listeria monocytogenes*, an intracellular pathogen, has also been used to develop a tumor vaccine [28]. Nevertheless, some cancer gene therapeutic vectors can still infect normal tissue [29,30]. *Bifidobacterium* is an obligate anaerobe that selectively localizes and proliferates within hypoxic regions of tumors as a non-pathogenic bacterium. It has been considered an alternative strategy in tumor therapy for its acknowledged recognized safety [19,31,32,33,34,35,36].

Our previous studies have confirmed that *Bifidobacterium infantis* (BF)-TK/GCV targeted solid tumors and inhibited the growth of gastric cancer (GC) and several other cancer cell lines through activating extrinsic and intrinsic apoptosis pathways [16,17,19,37,38,39]. In addition, we detected a significant decrease in CD146 expression after BF-TK/GCV treatment [39]. CD146 is reported to contribute to tumor metastasis and invasion [40,41]. So far, it remains unclear how many molecules are involved in inhibiting tumor cell metastasis after BF-TK/GCV treatment. This study aimed to decipher the mechanism of BF-TK/GCV on inhibiting tumor metastasis.

GC is one of the most common malignant tumors and the fourth leading cause of cancer death worldwide [42]. Despite the gradually declining mortality, GC still burdens many countries in East Asia [43]. Therefore, we comprehensively identified and quantified protein expression profiling in MKN–45 cells by BF-TK/GCV treatment using tandem mass tags (TMTs)-based quantitative proteomics and verified the expression of metastasis-related proteins by immunohistochemistry (IHC) in this study.

## 2. Results

### 2.1. Identification of Differentially Expressed Proteins (DEPs)

To better understand the mechanism by which BF-TK/GCV inhibited tumor cell metastasis occurs in GC, MKN−45 cells were treated with PBS, BF-TK, BF/GCV, and BF-TK/GCV, respectively. Then, the quantitative proteomic profiling was detected by 10-plex TMT. The workflow of the TMT-based proteomic analysis is demonstrated in Figure 1A (Figure 1).

Volcano plot filtering was used to identify DEPs with >1.2-fold changes and *p* values < 0.05. In the BF-TK/GCV/BF-TK group, 102 proteins were upregulated and 57 proteins were downregulated (Figure 2A). In the BF-TK/GCV/BF/GCV groups, 32 proteins were upregulated and 40 proteins were downregulated (Figure 2A).

Venn diagrams showed that eight and three proteins were observed with similar regulation between the BF-TK/GCV/BF-TK and BF-TK/GCV/BF/GCV groups, which suggested that they were important proteins for the effect of BF-TK/GCV (Figure 2A). Further analysis of the DEPs by unsupervised hierarchical clustering classified these samples into three different cohorts, which also reflected the three distinct characteristics (Figure 2B).

### 2.2. Functional Enrichment Analysis

To gain insight into the biological classifications between two groups, the DEPs between groups BF-TK/GCV/BF-TK and BF-TK/GCV/BF/GCV were analyzed separately using R software based on the GO database. In the BF-TK/GCV/BF-TK group, 2594, 346, and 402 terms in the BP, CC, and MF were enriched, respectively. The top 10 enriched BP terms are displayed in Figure 3A. BP analysis revealed significant enrichment for regulating dendrite development and lysosomal transport. Meanwhile, 1337, 189, and 214 terms were enriched in the BP, CC, and MF in the BF-TK/GCV/BF/GCV groups. The top 10 enriched BP terms are shown in Figure 3B. BP category analysis indicated that most genes were related to response to nutrient levels and endoplasmic reticulum stress (Figure 3).

Using R software, KEGG pathway enrichment analysis was performed on the two groups (BF-TK/GCV/BF-TK and BF-TK/GCV/BF/GCV). The top 30 enriched pathways are shown in Figure 3C,D. In the BF-TK/GCV/BF-TK group, KEGG pathway enrichment revealed that the DEPs were primarily implicated in the p53 signaling pathway (TNFRSF10B and Sulforaphane (SFN)) and necroptosis (TNFRSF10B, histone H2A variant H2AX (H2AX), and ATP/ADP translocase (SLC25A4)). The KEGG pathway analysis in the BF-TK/GCV/BF/GCV groups revealed the DEPs mainly participated in apoptosis (BCL2 associated X (BAX), myeloid cell leukemia-1 (MCL1), and α1 tubulin (TUBA1A)). These pathways are associated with cell death consistent with our previous studies [39,41]. In addition, there were pathways such as the Wnt signaling pathway (frizzled class receptor 6 (FZD6), peroxisome proliferator-activated receptor delta (PPARD), and secreted frizzled-related protein 4 (SFRP4)), gap junction (adenylate cyclase 8 (ADCY8) and neural tissue-specific F-actin-binding protein I (Neurabin-1, PPP1R9A)), and cell adhesion molecules (v-set immunoregulatory receptor (VSIR) and leukocyte antigen c (HLA-C)) associated with tumor metastasis. Subsequently, the Venn network showed the relationships between the top 30 enriched pathways and genes in the BF-TK/GCV/BF-TK and BF-TK/GCV/BF/GCV groups (Figure 3E,F).

### 2.3. PPI Networks and Module Analysis

Further, PPI networks were constructed based on the STRING database to clarify protein interactions (Figure 4A,B). MCODE was used to find clusters (highly interconnected regions) in the network, and the cluster with the highest score is shown in Figure 4C. In the BF-TK/GCV/BF-TK group, the cluster was composed of downregulated proteins: AFM (afamin), GIG25 (SERPINA3), APCS (serum amyloid P component), HRG (histidine-rich glycoprotein), ALB (serum albumin), HPX (hemopexin), and AMBP (alpha(1)-microglobulin/bikunin precursor). Meanwhile, the MCC method of the CytoHubba plug-in was used to calculate the top five proteins as hub proteins. In the BF-TK/GCV/BF-TK group, the top five hub proteins were AMBP, HRG, APCS, HPX, and GIG25 (Figure 4D); some proteins (HRG, HPX, and GIG25) are positively associated with tumor metastasis [44,45,46]. Moreover, the PPI network indicated that PPARD from the Wnt signaling pathway directly interacted with CEBPB, and indirectly interacted with NF-κB1, mTOR, HIF−1α, VCAM1, and CXCL12 via CEBPB (Figure 4E).

### 2.4. BF-TK/GCV Inhibits Gastric Cancer Metastasis

The Wnt signaling pathway, gap junction, and cell adhesion molecules signaling pathways were related to tumor metastasis. The MKN−45 cells treated by BF-TK/GCV demonstrated significantly reduced migration and invasion capabilities compared to the BF-TK or BF/GCV treatment in transwell assays (Figure 5). Moreover, WB and IHC assays further independently validated the several metastasis-related proteins mentioned above (Figure 6). The expression of HIF−1α, mTOR, NF-κB1-p105, VCAM1, CEBPB, MMP13, beta-catenin, and CXCL12 in the BF-TK/GCV group were significantly decreased compared with the BF-TK or BF/GCV groups (Figure 6). Obviously, ATG16 was significantly increased in the BF-TK/GCV group (Figure 6A). However, p-CREB was upregulated in the BF-associated groups compared with the PBS or GCV group (Figure 6D).

### 2.5. Clinical Significance of HIF−1α and VCAM1

TCGA data were used to validate the clinical significance of these proteins. The results revealed that HIF−1α and VCAM1 were significantly upregulated in STAD with regional lymph nodes metastasis (N) compared with no regional lymph node metastasis (N0) (Figure 7A). Moreover, the results suggested that STAD patients with a higher expression of VCAM1 exhibited a poorer overall survival rate (*p* = 0.026). HIF−1α had a similar but not significant trend, probably because of the smaller number of patients (*p* = 0.064) (Figure 7B,C). Furthermore, the expression of HIF−1α was positively correlated with that of VCAM1 (R = 0.28, *p* = 9.7 × 10^−9^), suggesting that HIF−1α may play a similar role to VCAM1 in the progression of GC (Figure 7D). BF-TK/GCV treatment decreased the expression of HIF−1α, VCAM1, and other factors, indicating that BF-TK/GCV treatment could improve the future overall survival rate.

## 3. Discussion

Although great progress has been made in cancer treatment over the past decade, the effective prevention of tumor metastasis and invasion is still challenging for clinical treatment [1,47]. In this study, we found that BF-TK/GCV inhibited tumor metastasis, providing a new antitumor metastasis strategy.

As a model of this study, GC is the fourth leading cause of death and the fifth most common cancer [42]. Although systemic chemotherapy, radiotherapy, surgery, immunotherapy, and targeted therapy have been shown to be effective in GC treatment [48], the five-year survival rate of GC patients is still poor because of cancer recurrence due to metastasis [49]. Thus, the ability of BF-TK/GCV to inhibit GC metastasis provided a new method and hope for prolonging the life of GC patients by inhibiting tumor metastasis and invasion.

The delivery vector is crucial to the HSV-TK/GCV system. Some viral-mediated HSV-TK/GCV can inhibit the growth or metastasis of insulinoma, glioblastoma, and prostate cancer cells. Nevertheless, severe immune and inflammatory responses and high production costs limit the use of these viral vectors [15,50]. Based on our previous research, here, we further validated that BF-TK/GCV also inhibited tumor metastasis and invasion. BF is a nonpathogenic, intrinsic, and anaerobic bacterium that selectively localizes and proliferates in the central area of solid tumors after administration vein or intratumor injection. BF-TK/GCV inhibited the growth of tumors through multiple mechanisms in our previous studies [16,17,19,37,38,39,51]. In this study, we further confirmed that several metastasis-related proteins decreased expression after BF-TK/GCV treatment, consistent with our previous findings that BF-TK/GCV induced metastasis-related VEGF and CD146 downregulation [19,39].

Consistent with our previous findings, this study also proved that several cancer death-associated signaling pathways were enriched in the BF-TK/GCV group, including apoptosis, necroptosis, and p53 signaling pathways to exert antitumor effect (Figure 3C,D) [37,39]. Moreover, vacuolar protein sorting (VPS) genes encode proteins involved in vesicular trafficking. VPS52 has been reported to inhibit the viability and induced apoptosis of GC cells in vitro [52]. Otherwise, VPS41 restrained methuosis and autophagy in cancer [53]. Fortunately, VP41 was downregulated and VPS52 was upregulated by BF-TK/GCV treatment, implying inhibition of tumor viability and growth (Figure 3A).

In this study, HPX, HRG, and GIG25 were significantly downregulated in the BF-TK/GCV/BF-TK group. Overexpression of HRG (histidine-rich glycoprotein) inhibits cell proliferation and increases apoptosis in hepatocellular carcinoma [54]. Hemopexin (HPX) promotes the invasion and metastasis of pancreatic cancer and colorectal carcinoma cells [55]. GIG25 is a serpin family A member 3 (or SERPINA3), and overexpression of SERPINA3 promotes tumor invasion and migration, and epithelial–mesenchymal transition in triple-negative breast cancer cells [56]. PELO was upregulated by BF-TK/GCV treatment and enriched in the mRNA surveillance pathway. However, it was listed outside of the top 30 pathways (data not shown in Figure 2A). PELO negatively regulates cell migration and metastasis in vivo [56].The results implied that BF-TK/GCV inhibited tumor metastasis through these proteins.

We found that gap junction signaling pathways (ADCY8 and PPP1R9A) were upregulated and cell adhesion signaling pathways (VSIR and HLA-C) were downregulated, which is associated with tumor growth metastasis.

Bax (BCL-2 Associated X) activation leads to the release of apoptotic factor cytochrome C and consequently to cancer cell apoptosis [57]. BF-TK/GCV significantly upregulated BAX (Figure 3B). Combined with the conclusion of our previous study that BF can induce tumor cell apoptosis, BF-TK/GCV was a promising antitumor agent for clinical treatment [17,19,38,51].

The low expression of ADCY8 is correlated with poor overall survival and progression-free survival in lung adenocarcinoma [58]. Similarly, VSIR (V-set immunoregulatory receptor) is a negative immune checkpoint regulator that inhibits antitumor immune responses. Anti-VSIR antibody treatment significantly reduces the number of metastatic nodules in the livers of mouse models of PDAC with liver metastases [59]. Our research showed that BF-TK/GCV downregulated VSIR, indicating its antitumor metastasis mechanism.

The tumor microenvironment of metastasis (TMEM) proteins can be described as tumor suppressors or oncogenes [60]. It is reported that the overexpression of circTMEM59 suppresses cell growth, enhances the cell death, and represses the metastatic behaviors of colorectal cancer cells [61]. TMEM59 was upregulated by BF-TK/GCV treatment (Figure 3A).

The IHC assay confirmed several metastasis-related proteins were downregulated by BF-TK/GCV treatment (Figure 6C,D). It was reported that the reduction in the gap junction and HIF1α and CXCR4 expression significantly inhibited the metastasis of breast cancer cells [62]. Moreover, neural cell adhesion molecule (NCAM) knockdown inhibits the metastasis of human melanoma cells via the Src/Akt/mTOR/cofilin pathway [63].

HIF-1 is the major regulator of oxygen homeostasis and consists of HIF−1α and HIF-1β subunits. The activation of HIF−1α promotes ovarian cancer cell migration and omental metastasis [64]. In this study, HIF−1α was reduced significantly in the BF-TK/GCV group compared with the BF-TK or BF/GCV groups.

mTOR is a serine/threonine protein kinase with two types of protein complex: mTORC1 and mTORC2. Furthermore, it promotes cancer cell metastasis via miR-451 and suppresses glioma cell proliferation and invasion in vitro and in vivo via suppression of the mTOR/HIF−1α/VEGF signaling pathway by targeting CAB39 [65]. In this study, mTOR and HIF−1α were significantly reduced in the BF-TK/GCV group compared to the BF-TK or BF/GCV groups (Figure 6A,B).

NF-κB belongs to a family of structurally related eukaryotic transcription factors, and the family is divided into p105/p50 (NF-κB1), p100/p52 (NF-κB2), RelA (p65), c-Rel, and RelB. It was reported that NF-κB could positively regulate downstream HIF−1α and VEGF, and is associated with ovarian cancer metastasis [66]. In this study, NF-κB1-p105 (NF-κB1) was lower in the BF-TK/GCV group compared to the BF/GCV group.

VCAM1 is an important member of the immunoglobulin superfamily and promotes the invasion and metastasis of colorectal cancer by inducing transendothelial migration [67]. The downregulation of VCAM1 expression blocks breast cancer cell metastasis [68]. In this study, VCAM1 was decreased significantly in the BF-TK/GCV group compared with the BF-TK group.

CEBPB is a member of the transcription factor family of CEBP. A previous study found that STAT3 promoted melanoma metastasis by upregulating the expression of CEBPB family members [69]. In this study, CEBPB was significantly reduced in the BF-TK/GCV group compared to the BF-TK or BF/GCV groups.

CXCL12 is a ligand of CXCR4, and the activation of CXCL12/CXCR4 makes M2 polarized macrophages promote liver cancer metastasis by secreting VEGF [70]. The CXCL12/CXCR4 axis plays a pivotal role in tumor development, survival, angiogenesis, metastasis, and tumor microenvironment. For example, the CXCL12/CXCR4 axis activates the NF-κB signaling pathway resulting in the induction of migration, invasion, and EMT processes [71]. BF-TK/GCV downregulated CXCL12 and NF-κB1-p105, indicating its antitumor metastasis mechanism via the downregulation of the CXCL12/CXCR4 axis/NF-κB signaling pathway.

ATG16 plays a key role in autophagy and forms the Atg5-Atg12-Atg16 complex [72]. Autophagy inhibition strongly promotes metastasis to the lung in breast cancer models [73]. Endogenous TMEM59 interacts with ATG16L1 and mediates autophagy in response to *Staphylococcus aureus* infection [74]. It was suggested that BF-TK/GCV significantly upregulated TMEM59 and ATG16, indicating the potential of inhibiting tumor metastasis.

BF-TK/GCV inhibited GC cell metastasis by downregulating the mTOR/HIF−1α pathway and the CXCL12/CXCR4 axis/NF-κB signaling pathway. More specifically, BF-TK/GCV treatment reduced the expression of HIF−1α, mTOR, NF-κB1-p105, VCAM1, CEBPB, and CXCL12, confirmed by IHC assay or other metastasis-related proteins (HLA-C, HPX, HRG, VSIR, and GIG25) and confirmed by proteomics assay. To better understand the underlying mechanisms, the exact role of these proteins will need to be further explored in the future.

## 4. Material and Methods

### 4.1. Cell and Animal Treatment

MKN−45 (RRID: CVCL_0434) cells were obtained from the Cell Bank of Type Culture Collection of the Chinese Academy of Sciences (Shanghai Institute of Cell Biology, Shanghai, China) and have been authenticated in the past three years. MKN−45 cells were maintained in a complete growth medium, RPMI 1640 medium with 10% fetal bovine serum, and all experiments were performed with mycoplasma-free cells. The cells were cultured in 100 mm culture dishes in a humidified, mixed environment of 37 °C and 5% CO_2_. The BF-TK/GCV suicide gene therapeutic system was constructed as described previously [17,19]. MKN−45 cells were treated with PBS, BF-TK, BF/GCV, or BF-TK/GCV for 48 h (GCV, 167 µg/mL), respectively.

BALB/c nude mice (male, 3–4 weeks, 20 g/mouse, n = 6) were housed at the Laboratory Animal Center of Chongqing Medical University (Chongqing, China). A mouse model of xenograft tumor was established by injecting MKN−45 cells (1.0 × 10^8^ cells/mouse) subcutaneously. The appearance of a visible tumor with a diameter greater than 0.5 mm proved that the tumor model was successful. Treatment was initiated when the tumor was 1.5 mm–2.0 mm in diameter. Each group was directly given PBS, BF-TK, BF/GCV, or BF-TK/GCV through intratumoral injections (BF or BF-TK was 1.0 × 10^6^ cells/tumor) on day 1 and day 4. GCV (5.0 mg/kg) was intraperitoneal injected daily for 7 days after BF and BF-TK administration. All mice were sacrificed on day 8 of GCV injection, and the subcutaneous tumors were completely excised. All mice were killed by sodium pentobarbital anesthesia for cervical dislocation.

This study was performed according to the guidelines for reporting in vivo experimental studies in animals and the study is reported in accordance with ARRIVE guidelines. The protocol was approved by the Committee on the Ethics of Animal Experiments at the Chongqing Medical University (SYXK2012-0001). All surgery was performed under sodium pentobarbital anesthesia, and all efforts were made to minimize suffering.

### 4.2. Total Protein Extraction

Total protein was extracted from the tumors. Cells were suspended in lysis buffer (1.0% sodium deoxycholate (SDS), 8 M urea) with appropriate protease inhibitors to inhibit protease activity. The mixture was allowed to settle at 4 °C for 30 min, vortexed every 5 min, and treated by ultrasound at 40 kHz and 40 W for 2 min. After centrifugation at 16,000× *g* at 4 °C for 30 min, the concentration of protein supernatant was determined by the bicinchoninic acid (BCA) method by BCA Protein Assay Kit (Pierce, Thermo, Rockford, IL, USA). Protein quantification was performed according to the kit protocol.

### 4.3. Liquid Chromatography-Tandem Mass Spectrometry (LC-MS/MS) Analysis

Protein digestion was performed according to the standard procedure, and the resulting peptide mixture was labeled using the 10-plex TMT reagent (Thermo Fisher, Art. No. 90111, Waltham, MA, USA) according to the manufacturer’s instructions [75]. Then, the pooled samples were fractionated into fractions by ACQUITY Ultra Performance liquid chromatography (Waters, Milford, MA, USA) with ACQUITY UPLC BEH C18 Column (1.7 µm, 2.1 mm × 150 mm, Waters, Milford, MA, USA) to increase proteomic depth. Finally, labeled peptides were analyzed by online nanoflow liquid chromatography-tandem mass spectrometry performed on a 9RKFSG2_NCS-3500R system (Thermo, Waltham, MA, USA) connected to a QExactive Plus quadrupole orbitrap mass spectrometer (Thermo, Waltham, MA, USA) through a nanoelectrospray ion source.

### 4.4. Proteomics Data Analysis

The RAW data were analyzed using Proteome Discoverer (Thermo Scientific, version 2.2, Waltham, MA, USA) against the uniprot-proteome_UP000005640-Homo sapiens (Human) [9606]-74468s-20190823.fasta database. The MS/MS search criteria were as follows: mass tolerance of 10 ppm for MS and 0.02 Da for MS/MS tolerance, trypsin as the enzyme with 2 missed cleavages allowed, carbamidomethylation of cysteine and the TMT of N-terminus and lysine side chains of peptides as fixed modification, and methionine oxidation as dynamic modifications, respectively. The false discovery rate (FDR) of peptide identification was set as FDR ≤ 0.01. A minimum of one unique peptide identification was used to support protein identification. The absolute value of fold change (FC) > 1.20 or < 0.83 and *p*-value < 0.05 were adopted as criteria for determining the significance of differential expression of a particular protein [76]. The subsequent data used for bioinformatics analysis were the RAW data processed by the difference analysis of BF/GCV vs. PBS, BF-TK vs. PBS, and BF-TK/GCV vs. PBS, so the PBS group data would no longer appear separately in the subsequent analysis.

### 4.5. Functional, Network, and Clinical Significance Analysis

The R (Version 4.1.2, https://www.r-project.org (accessed on 13 April 2022)) was used for the enrichment of biological processes (BP), molecular functions (MF), and cellular components (CC) based on Gene Ontology (GO) terms. Kyoto encyclopedia of genes and genomes (KEGG) analysis was used to identify enriched pathways [77]. A Venn network visualized the relationship between the top 30 enrichment pathways and related genes (Evenn, http://www.ehbio.com/test/venn/ (accessed on 13 April 2022) [78]). Protein–protein interaction (PPI) was assessed using the Search Tool for the Retrieval of Interacting Genes (STRING, version 11.5, https://string-db.org/ (accessed on 30 June 2022)), and the results under the medium confidence (String score = 0.4) were visualized in Cytoscape (version 3.9.1, http://www.cytoscape.org/ (accessed on 2 July 2022) [79,80]). Moreover, the top 5 hub proteins were identified with the MCC method by cytohubba, a plug-in of Cytoscape software [81]. Furthermore, the highest score clusters were analyzed by MCODE plug-in [82]. The Cancer Genome Atlas-Stomach Adenocarcinoma (TCGA-STAD) RNA-seq gene expression data and clinical data were obtained from the TCGA Data Portal (https://portal.gdc.cancer.gov/ (accessed on 3 July 2022) [83]). GEPIA was used to analyze the survival rate and gene expression correlation of high and low expression genomes (http://gepia.cancer-pku.cn/ (accessed on 3 July 2022) [84,85]).

### 4.6. Transwell Assay

The polycarbonate membrane was coated in the transwell chamber with matrigel (BD Falcon, Bedford, MA, USA) for the invasion assay. MKN−45 cells (approximately 1 × 10^5^ in 200 μL serum-free medium) were transferred into the top chamber and added medium with serum in the bottom chamber for 24 h at 37 °C. After the cells on the top side of the membrane were removed, the cells were fixed at the bottom side with 4% paraformaldehyde (PFA) and stained with 0.1% crystal violet. Cells at the undersurface of the chamber were imaged and counted.

### 4.7. Western Blot Analysis

Proteins were taken from the “total protein extraction” section of this study. Protein quantitation was performed by BCA protein assay reagent (Beyotime, Shanghai, China). Equal amounts of protein from the different groups were denatured in SDS sample buffer and separated on 8–10% polyacrylamide-SDS gel based on the protein molecular weight. Proteins were transferred to a polyvinylidene difluoride membrane. The antibodies to GAPDH (1:2000; D190090, Sangon Biotech, Shanghai, China), HIF−1α (hypoxia-inducible factor 1 subunit alpha, 1:2000; Cat# 66730, Proteintech, Wuhan, China), MMP13 (matrix-metalloproteinase-13, 1:2000; D198905, Sangon Biotech, Shanghai, China), beta-catenin (1:2000; C650043, Sangon, Shanghai, China), and ATG16 (1:200; D194517, Sangon Biotech, Shanghai, China) were used to detect the target proteins, followed by incubation with a secondary antibody conjugated with horseradish peroxidase. The proteins of interest were detected using the SuperSignal West Pico Chemiluminescent Substrate kit (Pierce, Thermo, Rockford, IL, USA). BIO RAD ChemiDoc™ Touch Imaging System (USA), GAPDH was used as a loading control, and the gray values of protein bands were measured using ImageJ software (Win 64-bit, National Institutes of Health, Bethesda, MD, USA).

### 4.8. Immunohistochemistry (IHC) Analysis

IHC analysis was performed to detect the expression of HIF−1α, VEGFA (vascular endothelial growth factor A), mechanistic target of rapamycin kinase (mTOR), nuclear factor kappa B subunit 1 (NF-κB1)-p105, the vascular cell adhesion molecule 1 (VCAM1), CCAAT enhancer binding protein beta (CEBPB), p-CREB (phospho-cAMP response element binding), CXCR4 (C-X-C chemokine receptor type 4), and c-x-c motif chemokine ligand 12 (CXCL12) in MKN−45 tumor xenograft tissues treated by PBS, BF-TK, BF/GCV, and BF-TK/GCV, respectively (with three replicates). Retrieved tissues were fixed, decalcified in 10% formalin, and embedded in paraffin 24 h post-treatment. The fixed GC tissues of MKN−45 were blocked and incubated with HIF−1α (1:200; D122477, Sangon Biotech, Shanghai, China), mTOR antibody (1:200; D160640, Sangon Biotech, Shanghai, China), NF-κB1 (1:200; D120133, Sangon Biotech, Shanghai, China), VCAM1 (1:200; D123530, Sangon Biotech, Shanghai, China), CEBPB (1:200; D155298, Sangon Biotech, Shanghai, China), p-CREB (1:200; D151216, Sangon Biotech, Shanghai, China), CXCR4 (1:200; D162693, Sangon Biotech, Shanghai, China), and CXCL12 (1:200; D161112, Sangon Biotech, Shanghai, China), respectively. After being washed, tissues were incubated with biotin-labeled secondary antibody for 30 min, followed by Streptavidin-HRP conjugate for 20 min at RT. The presence of the expected protein was visualized by DAB staining and examined under a microscope. Stains with control IgG were used as negative controls.

### 4.9. Statistical Analysis

Statistical analyses were performed using SPSS (Version 25.0, IBM, Chicago, IL, USA). One-way ANOVA was applied for the comparisons among multiple independent variables. The LSD Method was used for multiple pairwise comparisons. An independent sample *t*-test was used for two sample comparisons. The probability level at which the difference was considered significant was *p* < 0.05.

## 5. Conclusions

BF-TK/GCV inhibited tumor metastasis, which deepened and expanded understanding of the antitumor mechanism of BF-TK/GCV. The result is an important supplement to our previous research on the antitumor molecular mechanism of BF-TK/GCV [17,19,51]. It provides valuable data for the further study of BF-TK/GCV antitumor gene therapy.

## Figures and Tables

**Figure 1 ijms-24-11721-f001:**
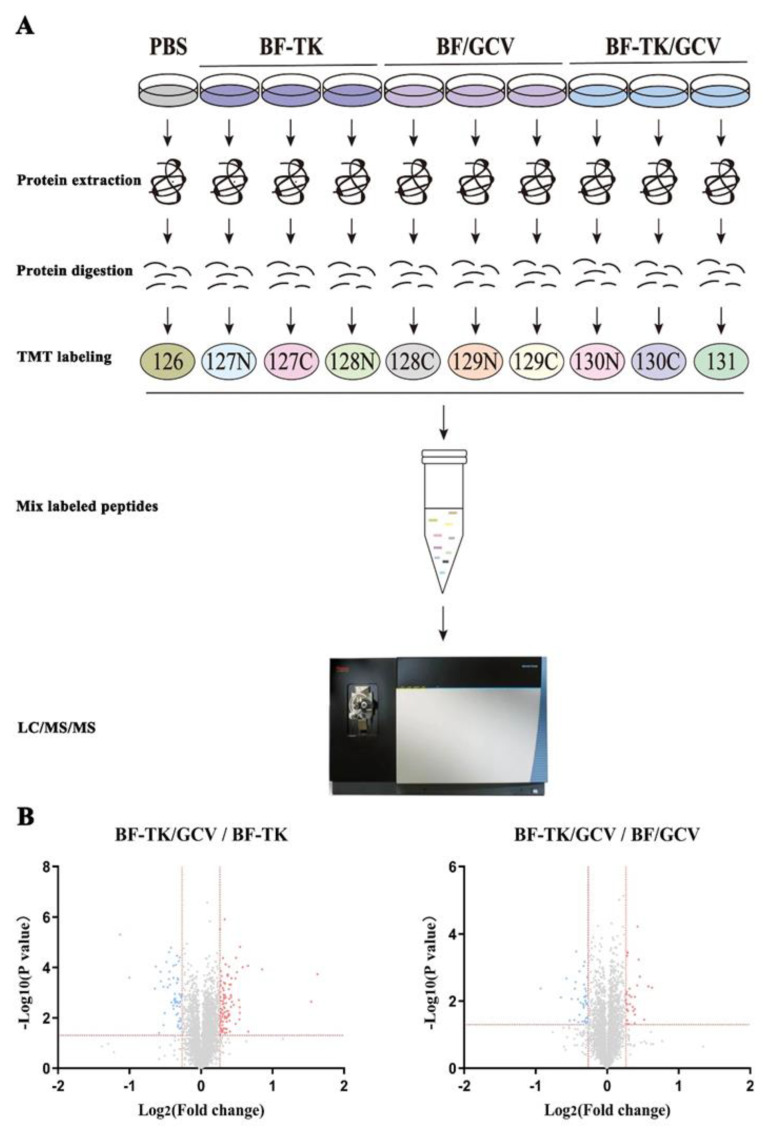
Comparative proteomic analysis of MKN−45 cells treated by BF-TK/GCV. (**A**) Schematic illustration of the TMT-based quantitative proteomic workflow. MKN−45 cells were treated with PBS, BF-TK, BF/GCV, or BF-TK/GCV for 48 h and were subjected to 10-plex TMT labeling. Labeled peptides are analyzed by online nanoflow liquid chromatography-tandem mass spectrometry. (**B**) Identification of differentially expressed proteins in BF-TK/GCV/BF-TK and BF-TK/GCV/BF/GCV groups. Low and high relative expressions are indicated in blue and red, respectively. Vertical red line represents 1.2 fold change, horizontal red line represents a *p* value of 0.05.

**Figure 2 ijms-24-11721-f002:**
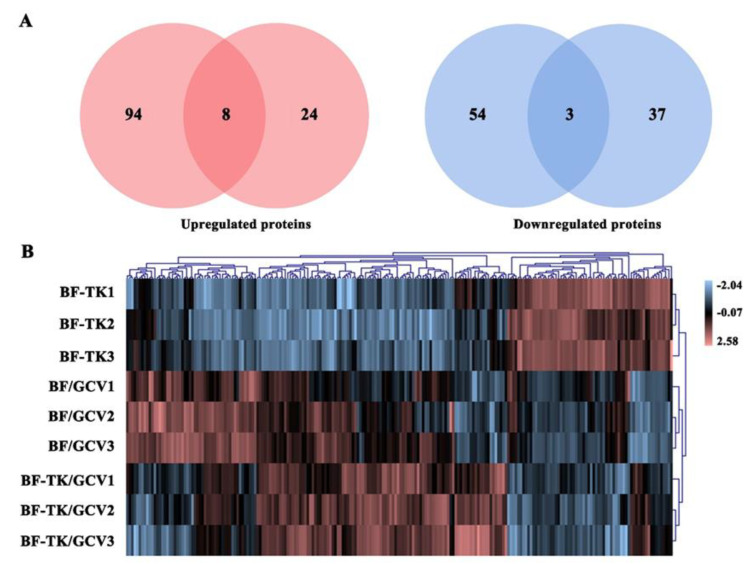
Expression analysis of the identified differentially expressed proteins. (**A**) Venn diagrams display the overlapping differential protein number among the BF-TK/GCV/BF-TK and BF-TK/GCV/BF/GCV groups. (**B**) Clustering of differential proteins from the three groups using a visualized heatmap. Pink signifies high relative expression and blue signifies low relative expression. Various color intensities indicate the expression levels, and a log2 scale is used in the color bar.

**Figure 3 ijms-24-11721-f003:**
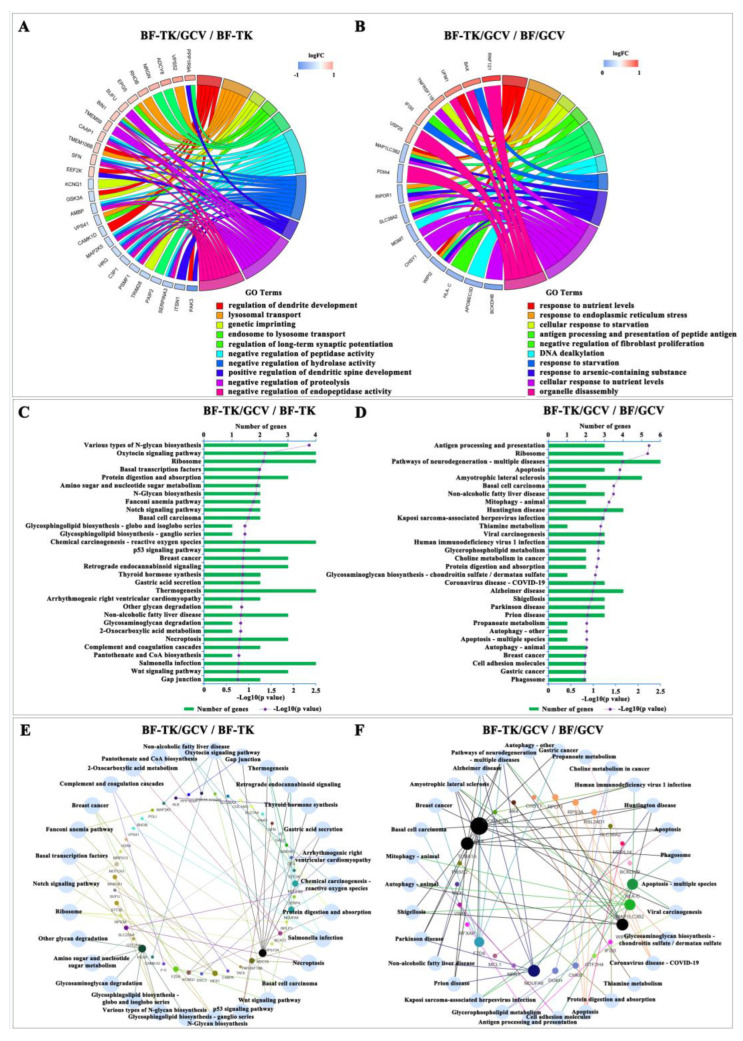
Gene Ontology, Kyoto Encyclopedia of Genes and Genomes pathway enrichment and Venn network analysis. Differentially expressed proteins from the (**A**) BF-TK/GCV/BF-TK and (**B**) BF-TK/GCV/BF/GCV groups are analyzed with the GO database, and the top 10 enriched BP terms are shown. The top 30 enriched KEGG pathways in the (**C**) BF-TK/GCV/BF-TK and (**D**) BF-TK/GCV/BF/GCV groups are displayed. Venn network of top 30 enriched pathways and proteins in the (**E**) BF-TK/GCV/BF-TK and (**F**) BF-TK/GCV/BF/GCV groups.

**Figure 4 ijms-24-11721-f004:**
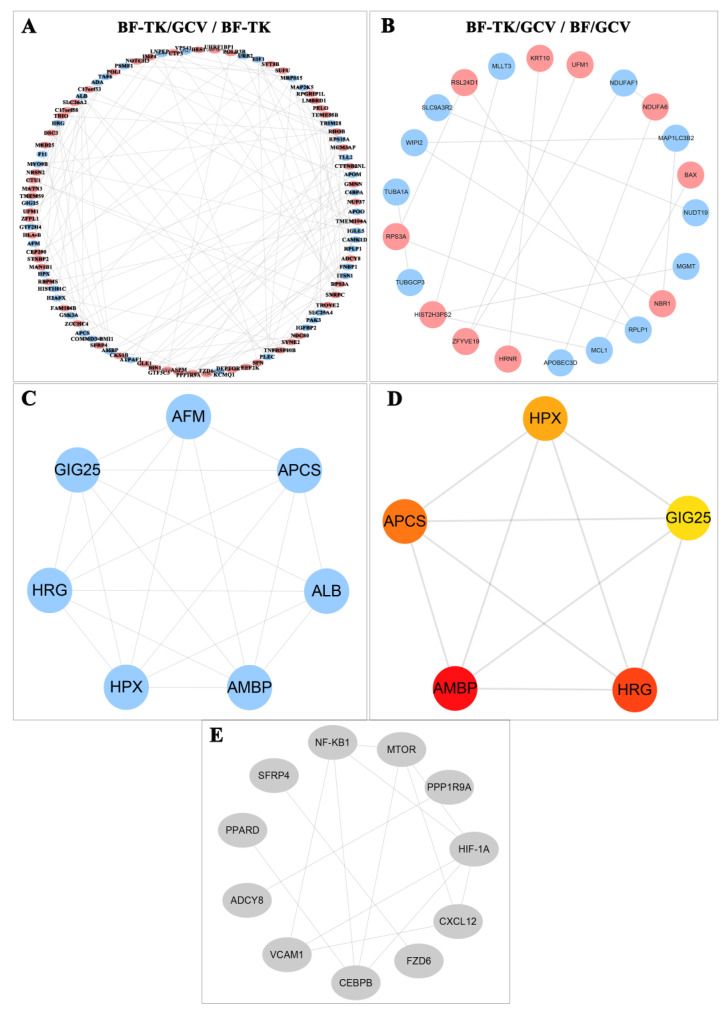
PPI network and subnetwork of the differentially expressed proteins. (**A**,**B**) Using the STRING online database, the differentially expressed proteins (upregulated proteins shown in red and downregulated proteins shown in blue) are filtered into a PPI network complex. The more interactions with other proteins, the more important this protein is. (**C**) Subnetworks are constructed using the MCODE plugin, and the subnetwork with the highest score is shown. (**D**) Subnetworks are constructed using the CytoHubba plugin; advanced ranking of hub proteins is represented by redder color. (**E**) A PPI network showing the interaction of PPARD with CEBPB and other factors indirectly.

**Figure 5 ijms-24-11721-f005:**
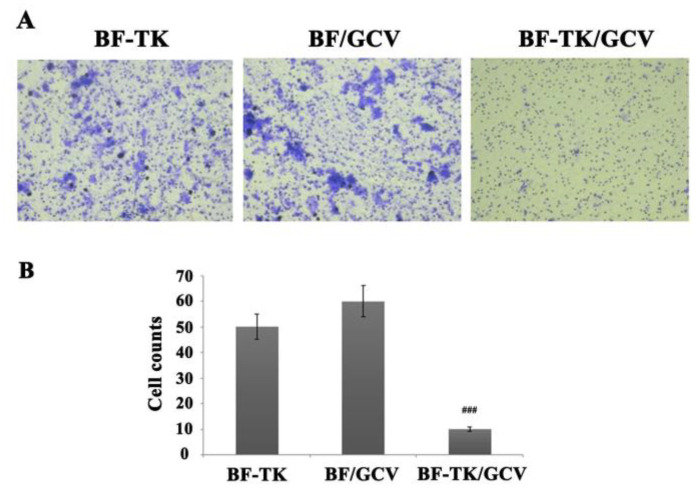
Transwell assay of metastasis-associated proteins. (**A**) Representative images of transwell assay and quantification of the number of cells that migrate or invade through the basement membrane. BF-TK/GCV significantly inhibited cell migration. (**B**) Quantification of transwell assay in different treatments. ### *p* < 0.001.

**Figure 6 ijms-24-11721-f006:**
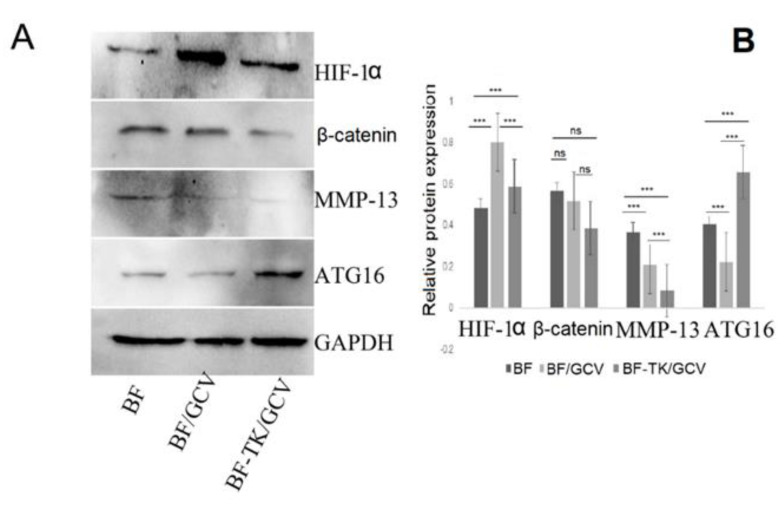
Western blot and IHC analysis of some metastasis-associated proteins. (**A**,**B**) HIF−1α, beta-catenin, and MMP13 were downregulated and ATG16 was upregulated by BF-TK/GCV treatment. ns, not significant; *** *p* < 0.001; (**C**) Representative tissue sections of CEBPB, CXCL12, CXCR4, HIF−1α, mTOR, NF-κB1-p105, p-CREB, VCAM1, and VEGFA in tumor tissues of different groups (200×). (**D**) Quantification of protein IHC staining; Data represent mean ± SD of three independent experiments. * *p* < 0.05 vs. BF-TK group; ** *p* < 0.01 vs. BF-TK group; *** *p* < 0.001 vs. BF-TK group; # *p* < 0.05 vs. BF/GCV group; ## *p* < 0.01 vs. BF/GCV group; ### *p* < 0.001 vs. BF/GCV group.

**Figure 7 ijms-24-11721-f007:**
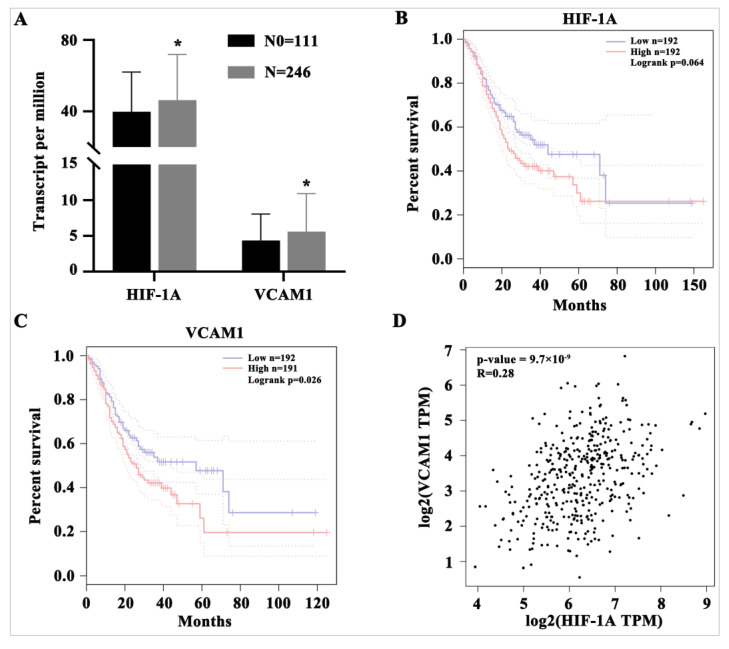
Clinical significance of HIF−1α and VCAM1. (**A**) Expression of HIF−1α and VCAM1 based on nodal metastasis status in TCGA-STAD. (**B**,**C**) Overall survival analysis of HIF−1α and VCAM1 in TCGA-STAD. (**D**) The correlation between HIF−1A and VCAM1 in TCGA-STAD. Data represent mean ± SD. * *p* < 0.05 vs. N0 group. N0: no regional lymph node metastasis.

## Data Availability

All data generated or analyzed during this study are included in this published article. The proteomics data can be freely and openly accessed at https://www.ebi.ac.uk/pride/ (accessed on 8 May 2023). Project accession: PXD042044.

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
