# Peer review of "Bifidobacterium infantis-Mediated Herpes Simplex Virus-TK/Ganciclovir Treatment Inhibits Cancer Metastasis in Mouse Model"

_ijms, 2023, doi:10.3390/ijms241411721_

Round 1

Reviewer 1 Report

Wang et al. demonstrated that BF_TK/GCV inhibits tumor metastasis providing a new anti-tumor metastasis strategy. Further it is found that BF-TK/GCV group associated with several death-associated signaling pathways, including apoptosis, necroptosis and p53 signaling pathways to exert anti-tumor effect.

Major concerns;

1. Authors should include the western blot for following genes including, HIF1A, NFkB1-p105, mTOR, VCAM1, CEBPB, CXCL12 to understand more precisely.

Minor concerns:

1. Figure 5, the quantification of western blots data -  the scale measurement of 'x' axis has to be labeled properly. 

2. Overall english language can be supervised to reach wider readers.

Needs Improvement

Author Response

Major concerns;

  1. Authors should include the western blot for following genes including, HIF1A, NFkB1-p105, mTOR, VCAM1, CEBPB, CXCL12 to understand more precisely.

Response: Thank you for your good advice. However, the protein samples were used

to LC-MS/MS at first. So the remaining samples were very limited. After we performed the trial test and tested HIF1α, β-Catenin and MMP13, the samples were run out. Considering that the samples vary widely from batch to batch, we did not prepare another batch protein samples. Instead, we used the same batch tissues to test the variation of NFkB1-p105, mTOR, VCAM1, CEBPB, and CXCL12 to understand the mechanisms. Although the forms were different, the results presented reflected the differences between each treatments. The most important thing was that the samples use in WB, IHC and LC-MS/MS were the same treatment.

Minor concerns:

  1. Figure 5, the quantification of western blots data -  the scale measurement of 'x' axis has to be labeled properly. 

Response: Be improved.

  1. Overall English language can be supervised to reach wider readers.

 Response: Be improved. Thank you so much.

Reviewer 2 Report

The present manuscript by Changdong Wang and colleagues studies the effect of Bifidobacterium infantis-mediated herpes simplex virus- 2 TK/ganciclovir treatment on gastric cancer in mouse model, and shows, BF-TK/GCV has antitumor properties. After quantitative proteomics and gene analysis authors found 159 and 72 differential expression proteins (DEPs) significantly changed in BF-TK/GCV/BF-TK and BF-TK/GCV/BF/GCV groups. Gene and genomics pathway analysis show enriched metastasis-related pathways such as gap junction and cell adhesion molecules pathways.

Overall, the work is ok, there are some issues needs to be fixed before it is accepted for the publication. Manuscript can be accepted after correction of following major concerns.

1.     The methods for animal treatments needs more clarity, please add few lines about tumor establishment. What were the markers/criteria used to establish the tumor growth? (i.e., Tumor measurement?), what was the criteria of starting the treatments?

2.     Is there any specific reason to use male mice?

3.     How much time it took to develop the tumor?

4.     Why only cervical dislocation was used to euthanize the mice, was any anesthesia given before cervical dislocation. was this the requirement of the experiment because anesthesia will not interfere with any tumor markers either molecular or biological.

5.     Please add a line in “Total protein extraction” section that “total protein was extracted from the tumors” As the protocol mention in “cell and animal treatment” section.

6.     “Liquid chromatography-tandem mass spectrometry (LC-MS/MS) analysis” section it is mentioned “Protein digestion was performed according to the standard procedure”. Please mention in brief how it was done or add the ref where you described the method.

7.     Results- for volcano plots figure 2A- the color representation of “, please remove the color box from the “upregulated or downregulated’ proteins, as all are same color (either pink or blue). From the figure legend of figure 2 please correct the red in to pink as there is no red color in heatmap, (line 214).

8.     IHC images are on low magnification and very small, hard to reach to the conclusion. All images should be brighter for better visualization. Please include either scale bars or magnification on the images or figure legend respectively.

9.     Western blots are not good quality, the quantification graphs are hazy should be replaced with better focused figures. Please include on the histograms if they are arbitrary units or gold changes etc. How the bands were calculated are they normalized with GAPDH? Mention clearly in the figure legend.

10. Figure 7 B and D, change HIF-1A to HIF-α, and change in figure legend accordingly.

11. Any abbreviation should be defined when it is mentioned first, like VSIR,

       Please defined this when used first (in line 354).

12.Throughout the discussion there are mistakes (coma, punctuation, sentences, e.g., from line 357-361, the whole sentence is very confusing), please read carefully and correct accordingly.

Author Response

  1. The methods for animal treatments needs more clarity, please add few lines about tumor establishment. What were the markers/criteria used to establish the tumor growth? (i.e., Tumor measurement?), what was the criteria of starting the treatments?

 Response: Be revised and the criteria was added.

  1. Is there any specific reason to use male mice?

Response: For one thing, male mice are cheaper. On the other hand, there will be no pregnancy and micebirth during the experiment, otherwise, the processing process is complicated.

  1. How much time it took to develop the tumor?

Response: For 3-4 weeks. The tumor formation time of different cell lines is different.

  1. Why only cervical dislocation was used to euthanize the mice, was any anesthesia given before cervical dislocation. was this the requirement of the experiment because anesthesia will not interfere with any tumor markers either molecular or biological.

   Response: 2,2, 2-tribromoethanol anesthesia was administered before cervical dislocation.

  1. Please add a line in “Total protein extraction” section that “total protein was extracted from the tumors” As the protocol mention in “cell and animal treatment” section.

Response: Be added. Thank you so much.

  1. “Liquid chromatography-tandem mass spectrometry (LC-MS/MS) analysis” section it is mentioned “Protein digestion was performed according to the standard procedure”. Please mention in brief how it was done or add the ref where you described the method.

Response: The protein extraction and digestion according to the manufacturer’s instructions (Thermo Fisher, Art. No.90111). a reference was add [44].

  1. Results- for volcano plots figure 2A- the color representation of “, please remove the color box from the “upregulated or downregulated’ proteins, as all are same color (either pink or blue). From the figure legend of figure 2 please correct the red in to pink as there is no red color in heatmap, (line 214).

Response: Thank you so much. Be revised.

  1. IHC images are on low magnification and very small, hard to reach to the conclusion. All images should be brighter for better visualization. Please include either scale bars or magnification on the images or figure legend respectively.

Response: Magnification  (200×, n=3) were added in figure legend. Thank you so much.

  1. Western blots are not good quality, the quantification graphs are hazy should be replaced with better focused figures. Please include on the histograms if they are arbitrary units or gold changes etc. How the bands were calculated are they normalized with GAPDH? Mention clearly in the figure legend.

Response: The WB was used the remaining protein samples after performed LC-MS/MS at first. So the remaining samples were very limited. After we performed the trial test and tested HIF1α, β-Catenin and MMP13, the samples were run out. Considering that the samples vary widely from batch to batch, we did not prepare another batch protein samples. So we are no anything to re-performed the WB to get a good quality figures now. However, the quantification graphs were clarified.

  1. Figure 7 B and D, change HIF-1A to HIF-α, and change in figure legend accordingly.

Response: Thank you so much. Be revised.

  1. Any abbreviation should be defined when it is mentioned first, like VSIR, Please defined this when used first (in line 354).

Response: Thank you so much. Be revised completely.

12.Throughout the discussion there are mistakes (coma, punctuation, sentences, e.g., from line 357-361, the whole sentence is very confusing), please read carefully and correct accordingly.

Response: Thank you so much. The sentence was deleted.

Round 2

Reviewer 1 Report

The authors addressed all of the queries.